# Disparities in Hepatitis B Vaccine Coverage by Race/Ethnicity: The National Health and Nutrition Examination Survey (NHANES) 2015–2016

**DOI:** 10.3390/diseases8020010

**Published:** 2020-04-16

**Authors:** Azad R. Bhuiyan, Nusrat Kabir, Amal K. Mitra, Oluwabunmi Ogungbe, Marinelle Payton

**Affiliations:** 1Department of Epidemiology and Biostatistics, School of Public Health, Jackson State University, Jackson, MS 39213, USA; nusrat.kabir@students.jsums.edu (N.K.); amal.k.mitra@jsums.edu (A.K.M.); marinelle.payton@jsums.edu (M.P.); 2School of Nursing, Johns Hopkins University, 525 N. Wolfe Street, Baltimore, MD 21205, USA; oogungb3@jhu.edu

**Keywords:** hepatitis B virus, HBV vaccine, race/ethnicity, health disparity, NHANES data

## Abstract

Hepatitis B virus (HBV) infection is the most common form of viral hepatitis and remains a global public health problem, even though the HBV vaccine is available. HBV leads to chronic liver disease, including cirrhosis, liver cancer, and death. This study aimed to identify disparities in HBV vaccine coverage with the serological test by race/ethnicity, adjusting for gender and age. In this study, 5735 adult participants were included, obtaining data from the National Health and Nutrition Examination Survey (NHANES), 2015–2016. Proc survey frequency, bivariate- and multivariate logistic regression in the weighted sample were performed due to the complex survey design of NHANES. Data were analyzed using SAS, version 9.2.4. The overall prevalence of HBV vaccine coverage was only 23.3% (95% CI: 20.7%, 25.9%). In a multivariate logistic regression model, data showed that Mexican Americans (OR 0.57, 95% CI: 0.38, 0.86) and African Americans (OR 0.70, 95% CI 0.56, 0.84) had lower vaccine coverage compared to Whites. Females had (OR 1.55, 95% CI: 1.30, 1.85) higher vaccine coverage compared to men. Older age groups (30–49 years) (OR 0.41, 95% CI: 0.32, 0.52) and age group ≥ 50 years (OR 0.18, 95% CI 0.14, 0.23) had lower vaccine coverage compared to younger adults aged 18–29 years.

## 1. Introduction

Hepatitis B virus (HBV) infection, a contagious disease, is a major public problem in the United States and around the world [1,2,3,4]. In the US in 2016, the prevalence of acute HBV infection was most common among injection drug users, those who had had sexual contact with a person with HBV infection, and those aged 30–49 years [5,6,7]. By contrast, the prevalence of chronic HBV infection was common among foreign-born individuals, especially people born in Asia, the Pacific Islands, and Africa, representing over half of the cases of chronic HBV infections [5,8]. Globally, in highly endemic areas, HBV infection was most commonly spread from mother to child at birth or through horizontal transmission (exposure to infected blood) and from an infected child to an uninfected child during the first 5 years of life [3]. In terms of morbidity and mortality, HBV infection affects 1.25 million persons in the US and 350 to 400 million persons worldwide. HBV infection causes 4000 to 5500 deaths in the US and one million deaths worldwide from its consequences of cirrhosis, liver failure, and hepatocellular carcinoma annually [1,2,3,4,5].

Although a safe and effective vaccine is available, three doses for children, and two doses for adults, HBV infections are highest among adults of ages 30–49 years [5]. The US 1991 public health strategy to eliminate transmission of HBV is still valid, in which the HBV vaccine is an essential prevention component [9]. Globally, in 1992, the World Health Organization recommended integrating the HBV vaccine into the national childhood immunization program by 1999 [10]. In the US, the HBV vaccine was first initiated in 1981 for screening all pregnant women and postexposure prophylaxis to infants born to infected women in 1988, a universal vaccine of children in 1992, and a routine vaccine of adolescents not vaccinated before in 1995 and unvaccinated children under age 19 in 1999 [11,12].

The National Viral Action Plan in the US, 2017–2020, emphasized the need to increase the percentage of persons aware of their HBV infection from 33% to 66%, to reduce the number of HBV-related deaths by 20%, and to increase the rate of HBV vaccination among health care workers to 90% [13]. According to the CDC report, only about 25% adults of the US who are at high risk for HBV infection get vaccinated [6]. Even among healthcare professionals, the vaccination coverage is only 69.5%, which is well below the Healthy People 2020 target [14]. According to several community-based studies, the vaccination rate remained low among American adults, and disparities in vaccine coverage exist [15,16,17,18,19,20,21,22]. Disparities also exist with testing for HBV infection. Among racial/ethnic minorities, more than half were not screened for HBV, and only half of those who tested positive had ever received treatment [4].

Therefore, this study aimed to investigate disparities associated with HBV vaccine coverage with serological tests by race/ethnicity from the current National Health and Nutrition Examination Survey (NHANES) 2015–2016, after adjusting for gender and age. We hypothesized that there is an association between differential HBV vaccine coverage measured by serological test and race/ethnicity, adjusting for gender and age.

## 2. Materials and Methods

### 2.1. Study Design

The NHANES 2015–2016 is a multistage, stratified, probability sampling design and is used to select a representative sample household of the US. It is conducted by the National Center for Health Statistics (NCHS). The survey combines interviews and physical examination to assess the health and nutritional status of non-institutionalized civilian adults and children in the US. A full description of the methodology for the NHANES can be obtained elsewhere [23]. The NHANES offers an initial interview component, which is usually conducted at the household level of each participant to collect sociodemographic information and medical histories. Physical examination, dietary interview, anthropometry, and other tests and procedures are conducted by highly trained medical personnel in mobile examination centers, which provide a standardized and controlled environment for clinical examinations and tests [24].

### 2.2. IRB Approval

The NCHS Research Ethics Review Board (ERB) approved the NHANES 2015–2016 survey (#2011-17). All participants gave their informed written consent for their participation in the survey.

### 2.3. Population

In our study, 5735 adult participants were included from the NHANES 2015–2016 survey. We included participants of all race/ethnicity categories, and adults (18 years and older), and excluded children and adolescents. Race/ethnicity of NHANES participants was categorized as follows: Mexican American, other Hispanic, White, African American, Asian, other races, and persons identified having more than one race. The age of participants was divided into three categories: 18–30 years, 30–49 years, and ≥ 50 years.

### 2.4. Measurements

HBV vaccine coverage was based on serological test measurements. The serologic test measures HBV-specific antigens and antibodies from a serum sample of participants. In the 2015–2016 NHANES laboratory tests, the anti-HBc assay, anti-HBs, and the HBsAg were performed. The hepatitis B core antibody tests were performed on all examined adult participants. By contrast, the hepatitis B surface antigen was done only when the hepatitis B core antibody test was positive. Participants whose results were positive on the hepatitis B surface antigen were coded positive. However, if either the test for surface antigen test or the hepatitis B core antibody were negative, participants were coded negative. A combination of markers from serological testing was used to determine the phases of HBV infection; susceptibility to the infection (if HBsAg—Negative, Anti-HBs —Negative, Anti-HBc—Negative), immunity due to HBV vaccination (if HBsAg—Negative, Anti-HBs—Positive, Anti-HBc—Negative), immunity due to natural infection (if HBsAg–Negative, Anti-HBs—positive, Anti-HBc—positive), and infected with HBV (if HBsAg—Positive, Anti-HBs—negative, Anti-HBc—positive [7,25].

### 2.5. Statistical Analysis

Data were analyzed using SAS, version 9.2.4. Proc survey frequency was performed due to the complex design of the NHANES survey. Weighted percentage with 95% confidence interval (CI) was reported to represent the US population. Logit confidence limits were computed for small percentages (less than 10%). Modified Rao–Scott Chi-square test was used to analyze if there were significant differences in hepatitis B vaccine coverage by age, gender, and race/ethnicity groups. Bivariate and multivariate survey logistic models were applied to determine independent predictors of vaccine coverage.

## 3. Results

In total, 240,414,647 populations were represented by the weighted sample (*n* = 5735). Of the total population, 63.3% were Whites, 9.2% were Mexican Americans, 6.4% were other Hispanics, 11.5% were African Americans, 5.8% Asians, and 3.7% were other races (including multiracial persons). For the gender and age group, 51.8% were female, 20.6% were 18–29 years old, 34.2% were 30–49 years old, and 45.2% were 50 years old and above (Table 1).

Table 2 displays the overall prevalence of those susceptible to HBV infection, which was 72.8% (95% CI: 70.2%, 75.5%) with the highest rate among Hispanics 79.9% (95% CI: 75.2%, 84.7%), followed by other Hispanics 74.0% (95% CI: 68.9%, 79.1%), and the lowest among Asians 45.9% (95% CI: 42.4%, 49.4%), females 75.7% 95% CI: 67.7%, 72.6%), and the older age group (50 and above) 81.3% (95% CI: 78.5%, 84.2%), followed by the 30–49 age group, 72.7% 95% CI: 69.0%, 76.3%). Immunity due to natural infection was highest among Asians 15.7% (95% CI: 10.2%, 21.2%) compared to all other races/ethnicities, ranging from 0.9% (95% CI: 0.5, 1.8*) in Hispanics to 9.6% (95% CI: 6.5%, 13.9%) in African Americans. With regard to age, 45.5% (95% CI: 40.3%, 50.7%) of younger adults (18–29 age group) had immunity due to natural infection, followed by 24.4% (95% CI: 20.5%, 28.4%) of 30–49 age group and then 12.8% (95% CI: 10.5%, 15.2%) of the older age group (50 and above). The overall prevalence of HBV vaccine coverage (immune due to vaccine) was only 23.3% (95% CI: 20.7%, 25.9%). By race/ethnicity, the vaccine coverage was 19.1% (95% CI: 14.3, 24.0) among Mexican Americans, 20.1% (95% CI: 14.5, 25.7) among other Hispanics, 23.3% (95% CI: 19.6, 27.0) among Whites, 22.5% (95% CI: 20.3, 24.8) among African Americans, 34.6% (95% CI: 29.4, 39.8) among Asians, and 26.1% (95% CI: 19.7, 32.6) among other races. Females had higher vaccine coverage than males (26.6%, 95% CI: 23.7–29.6 vs. 19.8%, 95% CI: 16.7, 22.9). Vaccine coverage was highest among younger adults (18–29 age group) at 45.5%, followed by the 30–49 age group at 24.4%, and lowest among the older age group (50 and above) at 12.8%.

The overall prevalence of HBV (acute or chronic) was low (0.3%, 95% CI: 0.2%, 0.5*%), with the highest prevalence being among Asians (3.8%, 95% CI: 2.9, 4.9*%).

In a multivariate logistic regression model (Table 3), data showed that Mexican Americans and African Americans (OR 0.57 with 95% CI: 0.38–0.86 and OR 0.70 with 95% CI 0.56–0.84, respectively) had lower vaccine coverage compared to Whites. Females (OR 1.55 with 95% CI: 1.30–1.85) had higher vaccine coverage compared to males. Older age groups (ages 30–49 and ≥50) had lower vaccine coverage compared to younger adults aged 18–29 (OR 0.41 with 95% CI: 0.32–0.52; and 0.18 with 95% CI 0.14–0.23, respectively).

In the subgroup analysis by age, Figure 1 shows that vaccine coverage is lowest among Hispanics, followed by African Americans aged 30–49.

## 4. Discussion

This study showed that HBV vaccine coverage was low across all races/ethnicities, including the Asian population, in US adults. HBV vaccine coverage was less than 35% across all races/ethnicities. Previous studies reported similar low HBV coverage as well as HBV testing in the US [4,16]. In our subgroup analysis of those aged 30–49, we found that Mexican Americans and other Hispanics had lower vaccine coverage within that age group. Based on our data, these ethnic groups were most susceptible to infection as they did not acquire immunity from the disease, compared to Asians and people of other races. Even though Asians had lower HBV vaccine coverage, 15% of people in that community became immune via natural infection. Moreover, the prevalence of HBV infection was 3% among the Asian population, which was consistent with a previous study [26]. Based on the CDC recommendation of routine testing of foreign-born people with a prevalence of ≥2% [27,28], urgent measures are needed to screen them to know their HBV status and treat them to reduce the rate of recurrence of infection, and incidence of cirrhosis of the liver, liver cancer, and mortality. Without intervention, the study showed that 25% of people with HBV die due to late complications of cirrhosis and liver cancer [29].

Our major aim was to investigate disparities associated with HBV vaccine coverage with serological tests by race/ethnicity adjusted for gender and age. In a multivariate model, after adjusting for gender and age, data showed that vaccine coverage was lower among Mexican Americans and African Americans. We also found that males had lower vaccine coverage than females. This finding is also consistent with a previous study from the NHANES [22,27]. Moreover, the prevalence of vaccine coverage was lower among the older age group compared to younger adults. The reason for the observed age difference is speculative. The universal vaccine initiative started in 1991, and this survey was completed in 2015–2016. Younger adults, who fall within that age range of 18–29 years, were more covered by the immunization program than adults [9]. In our subgroup analysis by age group, we also noticed that health disparity in vaccine coverage existed in the 30–49 age group. In that age group, vaccine coverage was lower among the Hispanic communities, followed by African Americans, which was also consistent with a previous study [22]. The current data provided evidence that hepatitis B vaccine coverage is still below the goal of Healthy People 2020 [14,28]. The prevalence of vaccinations among Hispanics has remained persistently low, as reported in the 2010 CDC vaccine coverage evaluation [29,30]. Our study also found that fewer Whites had received hepatitis B vaccination. This is in contrast to previous reports that have found a higher prevalence of vaccination in Whites compared to African Americans [14,22,24].

### 4.1. Strengths of the Study

A significant strength of this study is that the data were from a nationally representative sample of the US, and the study participants were included from all race/ethnicity groups, including the Asian population, in the NHANES. This Asian population was not adequately sampled in the earlier NHANES [31,32]. Another major strength is the use of serological tests to define vaccine coverage rather than relying on self-reported data on vaccine coverage. Recall bias was avoided via these methods.

### 4.2. Limitations of the Study

This was a cross-sectional study; hence, causality of the study finding cannot be claimed. We could not investigate prevalence and vaccine coverage among high-risk populations. Additionally, the findings of this study cannot be applied to institutionalized adult populations.

## 5. Conclusions

Despite certain limitations, the study showed a low prevalence of vaccine coverage among all races/ethnic groups, including Whites, in the general adult population. In a previous study, it was noted that HBV vaccine coverage was also low in high-risk adults [33]. Although the vaccine is safe and very effective against HBV infection, it has not yet reached the recommended level by Healthy People 2020. Therefore, the sequence of HBV infection is one of the major contributing factors for liver cancer death, whose rate is growing faster than the death rates for all other types of cancer [34]. Hence, the HBV vaccine is an essential prevention strategy to prevent liver cirrhosis, cancer, liver failure, and death. Actions are needed to improve vaccine coverage. Public health professionals should raise awareness of vaccine coverage among populations who did not receive the vaccine during their early life. Several studies suggest that sexually transmitted disease clinics, correction facilities, syringe exchange programs, soup kitchens, and drug treatment centers are useful targets for successful implementation of the HBV vaccine on a large scale [6,11].

## Figures and Tables

**Figure 1 diseases-08-00010-f001:**
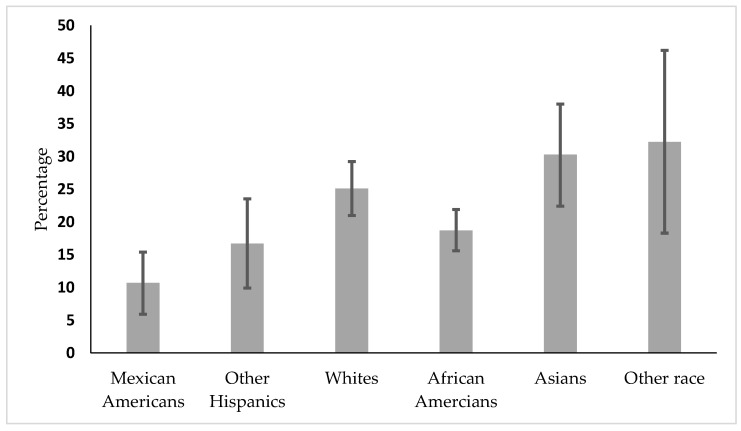
Vaccine coverage by race/ethnicity in the 30–49 age group.

**Table 1 diseases-08-00010-t001:** Demographic characteristics of the US adult population, National Health and Nutrition Examination Survey (NHANES), 2015–2016.

	Mexican Americans, *n* (%)	Other Hispanics, *n* (%)	Whites, *n* (%)	African Americans, *n* (%)	Asian Americans, *n* (%)	Others, *n* (%)	Overall
	1018 (9.2)	750 (6.4)	1839 (63.3)	1227 (11.5)	683 (5.8)	218 (3.7)	5735
Gender							
Male	475 (50.4)	322 (48.5)	929 (48.5)	576 (44.6)	343 (47.1)	114 (49.6)	2759 (48.2)
Female	543 (49.6)	428 (51.5)	910 (51.5)	651 (55.4)	340 (55.9)	104 (50.4)	2976 (51.8)
Age (years)							
18–29	238 (30.2)	135 (26.2)	324 (17.4)	270 (25.7)	166 (22.8)	59 (23.0)	1192 (20.6)
30–49	325 (40.2)	228 (41.9)	529 (31.3)	415 (36.2)	273 (42.5)	76 (33.2)	1846 (34.15)
≥50	455 (28.6)	387 (31.9)	986 (51.3)	542 (38.1)	244 (34.7)	83 (43.7)	2697 (45.2)

**Table 2 diseases-08-00010-t002:** Epidemiology of hepatitis B virus (HBV) infection status by race/ethnicity, gender, and age group.

Variables	Epidemiological Situation	Sample Frequency	Weighted Frequency	Weighted Percent	95% CI of Weighted Percent
Race/Ethnicity Hispanic	Susceptible to infection	799	16,665,192	79.9	75.2, 84.7
	Immune due to natural infection	13	187,389	0.9	0.5, 1.8 *
	Immune due to vaccine	150	398,4126	19.1	14.3, 24.0
	HBV infection	1	14,359	0.1	0.0, 0.5 *
	Total	963	20,851,067	100.0	
Other Hispanic	Susceptible to infection	519	10,384,106	74.0	68.9, 79.1
	Immune due to natural infection	47	827,351	5.9	4.4, 7.9 *
	Immune due to vaccine	113	2,824,185	20.1	14.5, 25.7
	HBV infection	0	-	-	-
	Total	679	14,035,642	100.0	
White	Susceptible to infection	1316	107,066,942	74.9	71.1, 78.7
	Immune due to natural infection	40	2,500,117	1.7	1.3, 2.4 *
	Immune due to vaccine	354	33,337,328	23.3	19.6, 27.0
	HBV infection	1	33,041	0.0	0.0, 0.2 *
	Total	1711	142,937,427	100.0	
African American	Susceptible to infection	725	16,198,029	67.3	63.5, 71.2
	Immune due to natural infection	111	2,302,969	9.6	6.5, 13.9
	Immune due to vaccine	216	5,421,330	22.5	20.3, 24.8
	HBV infection	6	129,616	0.5	0.3, 1.1 *
	Total	1058	24,051,943	100.0	
Asian	Susceptible to infection	262	5,400,762	45.9	42.4, 49.4
	Immune due to natural infection	92	1,843,881	15.7	10.2, 21.2
	Immune due to vaccine	193	4,068,564	34.6	29.4, 39.8
	HBV infection	24	448,139	3.8	2.9, 4.9 *
	Total	571	11,761,346	100.0	
Other race	Susceptible to infection	133	5,948,253	71.3	64.2, 78.5
	Immune due to natural infection	8	179,001	2.1	0.8, 5.3 *
	Immune due to vaccine	53	2,178,136	26.1	19.7, 32.6
	HBV infection	1	31,604	0.4	0.0, 3.5 *
	Total	195	8,336,994	100.0	
Total	Susceptible to infection	3754	161,663,283	72.8	70.2, 75.5
	Immune due to natural infection	311	7,840,708	3.5	2.9, 4.2 *
	Immune due to vaccine	1079	51,813,669	23.3	20.7, 25.9
	HBV infection	33	656,758	0.3	0.2, 0.5 *
	Total **	5177	221,974,419	100.0	
Gender					
Male	Susceptible to infection	1865	81,157,850	75.7	72.1, 79.2
	Immune due to natural infection	176	4,378,371	4.1	3.4, 4.9 *
	Immune due to vaccine	424	21,262,212	19.8	16.7, 22.9
	HBV infection	24	465,274	0.4	0.2, 0.9 *
	Total	2489	107,263,707	100.0	
Female	Susceptible to infection	1889	80,505,433	70.2	67.7, 72.6
	Immune due to natural infection	135	3,462,337	3.0	2.2, 4.2*
	Immune due to vaccine	655	30,551,458	26.6	23.7, 29.6
	HBV infection	9	191,484	0.2	0.1, 0.3 *
	Total	2688	114,710,712	100.0	
Age Group (years)					
18–29	Susceptible to infection	590	23,694,601	53.6	48.4, 58.7
	Immune due to natural infection	12	326,514	0.7	0.5, 1.2 *
	Immune due to vaccine	448	20,126,756	45.5	40.3, 50.7
	HBV infection	4	88,551	0.2	0.1, 0.4 *
	Total	1054	44,236,423	100.0	
30–49	Susceptible to infection	1213	55,490,499	72.7	69.0, 76.3
	Immune due to natural infection	77	1,936,718	2.5	1.7, 3.7 *
	Immune due to vaccine	383	18,658,891	24.4	20.5, 28.4
	HBV infection	13	264,293	0.3	0.2, 0.7 *
	Total	1686	76,350,401	100.0	
≥50	Susceptible to infection	1951	82,478,183	81.3	78.5, 84.2
	Immune due to natural infection	222	5,577,476	5.5	4.6, 6.6 *
	Immune due to vaccine	248	13,028,022	12.8	10.5, 15.2
	HBV infection	16	303,915	0.3	0.1, 0.6 *
	Total	2437	101,387,596	100.0	

* Logit confidence limits are computed for small percentages (less than 10%). ** 558 participants’ serological tests are undetectable. Susceptible to infection (if HBsAg—Negative, Anti-HBs—Negative, Anti-HBc—Negative); immunity due to HBV vaccination (if HBsAg—Negative, Anti-HBs—Positive, Anti-HBc—Negative); immunity due to natural infection (if HBsAg—Negative, Anti-HBs—Positive, Anti-HBc—Positive); infected with HBV (if HBsAg—Positive, Anti-HBs—Negative, Anti-HBc—Positive).

**Table 3 diseases-08-00010-t003:** Predictor of vaccine coverage by demographic variables.

	Bivariate Model	Multivariate Model
Variables	OR ratios with 95% CI	OR ratios with 95% CI
Race	-----	-----
White	Referent	Referent
Mexican American	0.79 (0.52–1.20)	0.57 (0.38–0.86)
Other Hispanic	0.80 (0.54–1.17)	0.61 (0.38–1.00)
African American	0.87 (0.74–1.02)	0.70 (0.56–0.84)
Asian	1.46 (1.11–1.93)	1.24 (0.95–1.64)
Other race	1.14 (0.75–1.74)	1.02 (0.62–1.66)
Gender	------	------
Male	Referent	Referent
Female	1.45 (1.21–1.72)	1.55 (1.30–1.85)
Age group	------	------
18–29	Referent	Referent
30–49	0.43 (0.34–0.54)	0.41 (0.32–0.52)
≥50	0.20 (0.15–0.26)	0.18 (0.14–0.23)

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
