# Peer review of "Disparities in Hepatitis B Vaccine Coverage by Race/Ethnicity: The National Health and Nutrition Examination Survey (NHANES) 2015–2016"

_diseases, 2020, doi:10.3390/diseases8020010_

Round 1

Reviewer 1 Report

General comment

The study reports a descriptive analysis on the hepatitis B vaccine coverage of a representative sample of the American population in relation to race/ethnicity, gender and age differences.

The study may have some points of interest, however the analysis and discussion of the results is very poor. The manuscript appears more like a brief communication of simple descriptive data than an original research. Although the data collected through the survey are many, unfortunately the Authors have limited themselves to a banal analysis of frequencies; furthermore, in the discussion section Authors merely repeat the results obtained without discussing them in the light of the available literature. Just to give an example, some interesting research has recently been published on the topic of vaccinations (also against hepatitis B) and gender differences

The reviewer believes that this version of the manuscript, without a thorough review of the results and discussion sections, is of very little interest to the scientific community.

In my opinion the manuscript must be rejected.

Author Response

Response to Reviewer 1

  • Introduction: Background information with relevant references are updated.
  • Research design: This is a cross-sectional study, using secondary data. These statements are added in the study design in line,  70-72.
  • Methods: Described more elaborately.
  • Results and Discussion: Revised extensively using the new information obtained from bivariate and multivariate logistic regression models to determine the independent predictor(s) of vaccine coverage. Point estimates with 95% CI were added after using proc survey frequency which represents the estimate of HBV coverage in the population.

Reviewer 2 Report

This is an interesting study exploring the impact of relevant socio-demographic characteristics on HBV vaccine coverage in a large and representative sample of US population. However, the manuscript would benefit from some revisions and additional analysis.

Abstract

Line 17: check language

Introduction

Line 42-43: sentence not clear, please revise Line 50-61: this paragraph does not seem to follow a clear logic. Please revise to provide the reader a clearer understanding of: HBV epidemiology; available diagnostics; vaccination strategy in the country.

Materials and methods

Line 88: check language Could the author explain why the selected cut-off was 20 years? This is unusual Line 93: Anti-HBs test should be included in the list as well. Line 104-107: definition of race. Please consider revising the sentence as it is unclear Line 110 vs 85: repetition

Results:

Table 2: would be useful to add a column for total study population Line 127-128: Consider revising the analytical approach to include individuals with HBV chronic infection or with resolved infection. This would allow a better overview of the epidemiological situation of the study population, i.e. susceptible individuals; non-susceptible individuals (vaccinated; resolved infection); chronically infected individuals as described in the methods section Figure 1: Consider re-organising the figure placing the bars progressively according to coverage level Analysis by gender, age and race could perhaps be better displayed in a table. Have the authors considered a multivariate analysis?

Discussion

“We also found that less than a quarter (24%) of the overall population had received at least three doses of hepatitis B vaccine” – according to data presented it is not possible to assess whether or not the individuals who have a vaccine-induced immunity have received the entire vaccination course. Please revise. Line 159: “…the New York city 160 population and Rocky mountain area..” no geographical analysis has been performed. The authors should consider including it in the study. Line 167: “…reflecting the highest prevalence of HBV infection among older age groups due to low vaccine coverage…”. This sentence is not supported by the data as prevalence of natural infection-derived immunity or chronic infection is not presented in the results section. Discussion could be implemented expanding more on reasons for disparities between race and age groups, including possible explanations for Asian people to have such higher vaccination level.

Author Response

Response to Reviewer 2

  • Abstract, Line 17: Revised as follows - 5,735 adult participants were included in this study from the National and Health Nutrition Examination Survey (NHANES), 2015-2016.
  • Introduction, Line 42-43: Updated with references.
  • Materials and Methods, Line 50-61: Modified as follows:

HBV vaccine has been available in the U.S. since 1981. It is the world's first (liver) cancer prevention vaccine and the first vaccine to prevent a sexually transmitted disease [10]. Adults who have had a chronic HBV infection since childhood develop hepatocellular carcinoma (HCC) at a rate of 5% per decade, which is 100-300 times the rate among uninfected people [11,12]. The test for the hepatitis B surface antibody (anti-HBc) indicates recovery and immunity from HBV. It also develops in a person who has been vaccinated against HBV [13]. The test for the total anti-HBc indicates previous or ongoing infection with HBV. The test for the HbsAg is also performed when the anti-HBc test is positive [14]. Only about 25% of the U.S. adults who are at high risk for HBV infection get vaccinated according to CDC recommendation [15]. The U.S. 1991 public health strategy to eliminate transmission of HBV is still valid, in which the HBV vaccine is a very important prevention component [16]. According to several community-based studies, the vaccination rate remained low among American adults, and disparities in vaccine coverage exist [17-21].

  • Line 88 has been modified.
  • Line 90: We changed the cutoff point for adults from 20 to 18 based on the fact that a cutoff point of 18 years is considered an adult in most of the states in the U.S.
  • Line 93: Anti-HBs is added.
  • Line 104-107: Race definition was modified.
  • Line 110 vs 85: The repetition was removed.
  • Results: A column for the total study population is added in Table 2, as suggested.
  • Line 127-128: Analytic approach was modified to include individuals with HBV infection and vaccine-induced immunity status. A new Table is included and graphs are removed.
  • Bivariate and multivariate logistic regression models were applied for independent predictor(s) of vaccine coverage using demographic variables.
  • Discussion:
  • We removed 3 doses of vaccine part from the statement.
  • The regional part was removed from the discussion section.
  • Line 167 removed.

Expanded the discussion part as suggested.

Reviewer 3 Report

Thank you for the opportunity to revise this interesting manuscript.

I have found the study design appropriate, and the methods and results clearly described.

As I believe that the study could be of interest to the readers, I suggest a few minor amendments before its publishing in the Journal. 

1) Even though the introduction section is well presented, epidemiological data on the HBV infections refer to bibliography references not recently published. I suggest the authors add more recent references to describe the world epidemiological panorama.

2) The Conclusions section is very generic. How can Public health professionals raise awareness of vaccine coverage among the general population? How can policymakers work on increase vaccine coverage through health insurance coverage? Some practical example of the strategies that can be implemented is needed.

I hope the authors will follow the suggestion because I believe that the study can be of interest to the readers.

Author Response

Response to Reviewer 3

  • The introduction part with references is updated.
  • Results section is extensively updated and incorporated in conclusion section.
  • In the revised version, we included three updated tables with bivariate and multivariate analyses, and one new figure to illustrate the vaccine coverage by race/ethnicity among 30-49 age group.
  • The conclusion section has been modified, as suggested.

Round 2

Reviewer 1 Report

General comment

The Reviewer carefully read the new version of the manuscript. Although the Authors added some new analysis, and implemented the tables, they failed to present and discuss their results adequately and of interest to the scientific community. From the very poor discussion, no original ideas emerge.

In my opinion, the new version of the manuscript has not significantly improved.

Author Response

Disparities in Hepatitis B Vaccine Coverage by Race/Ethnicity: National Health and Nutrition Examination Survey (NHANES) 2015-2016

Response to Reviewer 1

Round 2

Yes

Can be improved

Must be improved

Not applicable

Does the introduction provide sufficient background and include all relevant references?

( X)

(x)

( )

( )

Is the research design appropriate?

( X)

(x)

( )

( )

Are the methods adequately described?

(X )

(x)

( )

( )

Are the results clearly presented?

( )

( X)

(x)

( )

Are the conclusions supported by the results?

(X )

()

( )

( )

Comments and Suggestions for Authors

General comment

The Reviewer carefully read the new version of the manuscript. Although the Authors added some new analysis and implemented the tables, they failed to present and discuss their results adequately and of interest to the scientific community. From the very poor discussion, no original ideas emerge.

In my opinion, the new version of the manuscript has not significantly improved.

Responses

  • Introduction: Background information with relevant references are updated with new references in lines 30-90.
  • Research design: This is a cross-sectional study, using secondary data. These statements are added in the study design inline 66-70.
  • Methods: Described more elaborately.
  • Results: This section is updated with results from Table 2 in line from 121 to136
  • Discussion: Updated based on the findings. Please see lines 152-165. Our study findings were compared with the previous results. The original ideas emerge from the study were discussed. Our major aim was to investigate disparities associated with HBV vaccine coverage with serological tests by race/ethnicity adjusted for gender and age. In a multivariate model, after adjusting for gender and age, data showed that vaccine coverage was lower among Mexican Americans and African Americans. We also found that males had lower vaccine coverage than females. This finding is also consistent with a previous study from the NHANES survey. Moreover, the prevalence of vaccine coverage was lower among the older age group compared to younger adults. Please see inline 167-183 without mark up.
  • The conclusions have been revised (lines 195-206). Implications of this study have been highlighted as follows: Public health professionals should raise awareness of vaccine coverage among populations who did not receive the vaccine during their early life. Several studies suggest that sexually transmitted disease clinics, correction facilities, syringe exchange programs, soup kitchens, and drug treatment centers are useful targets for successful implementation of the HBV vaccine on a large scale.